# Blockade of Human α7 Nicotinic Acetylcholine Receptor by α-Conotoxin ImI Dendrimer: Insight from Computational Simulations

**DOI:** 10.3390/md17050303

**Published:** 2019-05-23

**Authors:** Xiaoxiao Xu, Jiazhen Liang, Zheyu Zhang, Tao Jiang, Rilei Yu

**Affiliations:** 1Key Laboratory of Marine Drugs, Chinese Ministry of Education, School of Medicine and Pharmacy, Ocean University of China, Qingdao 266003, China; 21170831073@stu.ouc.edu.cn (X.X.); 11180822010@stu.ouc.edu.cn (J.L.); zzy8617@stu.ouc.edu.cn (Z.Z.); jiangtao@ouc.edu.cn (T.J.); 2Laboratory for Marine Drugs and Bioproducts of Qingdao National Laboratory for Marine Science and Technology, Qingdao 266003, China; 3Innovation Center for Marine Drug Screening & Evaluation, Qingdao National Laboratory for Marine Science and Technology, Qingdao 266003, China

**Keywords:** 2×ImI-dendrimer, hα7 nAChR, linker, binding affinity, molecular dynamics simulation, MMGB/SA

## Abstract

Nicotinic acetylcholine receptors (nAChRs) are ligand-gated ion channels that are involved in fast synaptic transmission and mediated physiological activities in the nervous system. α-Conotoxin ImI exhibits subtype-specific blockade towards homomeric α7 and α9 receptors. In this study, we established a method to build a 2×ImI-dendrimer/h (human) α7 nAChR model, and based on this model, we systematically investigated the molecular interactions between the 2×ImI-dendrimer and hα7 nAChR. Our results suggest that the 2×ImI-dendrimer possessed much stronger potency towards hα7 nAChR than the α-ImI monomer and demonstrated that the linker between α-ImI contributed to the potency of the 2×ImI-dendrimer by forming a stable hydrogen-bond network with hα7 nAChR. Overall, this study provides novel insights into the binding mechanism of α-ImI dendrimer to hα7 nAChR, and the methodology reported here opens an avenue for the design of more selective dendrimers with potential usage as drug/gene carriers, macromolecular drugs, and molecular probes.

## 1. Introduction

Nicotinic acetylcholine receptors (nAChRs) are cation-selective pentameric ligand-gated ion channels that belong to the Cys-loop receptor family, which also includes γ-aminobutyric acid type A (GABAA), glycine, and serotonin (5-HT3) receptors [1,2]. It consists of three parts, an extracellular domain, a transmembrane domain, and an intracellular domain (Figure 1A). Up to now, more than 10 different neuronal nAChR subtypes have been discovered, including α7, α9α10, α3β2, and α4β2 [3]. Each subtype is closely related to a series of pathophysiological functions, such as Parkinson’s disease, Alzheimer’s disease, epilepsy, and so on [4,5,6]. For instance, disorders of α7 nAChR can cause schizophrenia and Alzheimer’s disease [7]. Therefore, nAChR inhibitors with high potency and low toxicity can be potentially developed into drugs for the treatment of these major diseases [8,9].

Conotoxins derived from the venom of *Conus* snails are pharmacologically valuable peptides with potency at various ion channels [10,11,12,13]. Most of the reported conotoxins targeting nAChRs are α-conotoxins, which belong to the most widely studied conotoxin class [12]. α-Conotoxin ImI (α-ImI) features a short α-helix and two disulfide bonds that link cysteines I–III and II–IV (Figure 1B). It comprises 12 residues and is C-terminal amidated (indicated by * in the sequence) (Figure 1C). α-ImI binds specifically to the human (h) α7 and α3β2 nAChR with IC_50_ of 595 nM and 40.8 nM, respectively [13,14]. Currently, α-conotoxins are receiving increased attention from researchers due to their potential medicinal value or potential use as molecular probes for neuropharmacology study.

Dendrimers with polyvalent structures, first developed in the 1980s [15,16], have been widely regarded as useful carriers for small molecular drugs and gene delivery [17,18]. By integrating the merits of dendrimers with bioactive peptides, peptide-decorated dendrimers (PDDs) have been extensively fabricated to generate vaccines [19,20,21,22,23,24], antiviral agents [25,26], and antitumor therapeutics [27,28,29,30,31,32] due to their multivalency effects. In particular, dendrimeric peptides have shown interesting properties for enhancing overall binding affinity and specificity compared with monovalent ligands. For instance, the α-ImI dimer showed substantially enhanced potency at hα7 nAChR [33], and in 3–5% *w*/*w* SPL7013 gels, a dendrimer with a polyanionic outer surface was effective in blocking vaginal transmission of simian/HIV-1 virus (SHIV) in macaques [34]. Despite its promising applications, rational design of dendrimeric peptides to achieve optimal bioactivity has proven to be challenging [35,36]. Hence, it is a prerequisite to establish a computational method for the design of dendrimeric peptides in silico prior to chemical synthesis in the laboratory. For this purpose, we used a computational modeling method to build models of α-ImI dendrimer bound with hα7 nAChR and calculated its binding energy. The simulation results shed light on the interaction mechanism between α-ImI dendrimer and hα7 nAChR.

## 2. Results and Discussion

### 2.1. Comparing the Binding Mode of α-ImI Monomer and α-ImI Dimer at hα7 nAChR

The α-ImI monomer and α-ImI dimer possessed similar binding modes at hα7 nAChR (Figure 2A,B and Appendix A). The critical residues conferring α-ImI specificity for hα7 nAChR were clustered at one site of α-ImI, which is consistent with a previous NMR structural study of α-ImI [37]. Comparing the conformations of α-ImI before and after molecular dynamics (MD) simulations, it was found that the conformations of D5, R7, and W10 were well maintained, while R11 demonstrated a slight orientation change (Figure 2A,B). Results from a previous mutagenesis study and computational modeling suggested that D5, R7, and W10 in α-ImI contribute most to the binding affinity of α-ImI [38,39], which is in line with the results from our modeling of α-ImI/hα7-nAChR monomers and 2×ImI-dendrimer/hα7-nAChR in the current study (Figure 2A,B). The overall results suggest that conformations for these critical residues in each α-ImI of the dimer were maintained after MD simulations, and the linker had minor effects on α-ImI binding to the hα7 nAChR.

### 2.2. Effects of the Linker to the Binding of 2×ImI-Dendrimer with hα7 nAChR

#### 2.2.1. Conformation of α-ImI in the 2×ImI-Dendrimer

Conformation of α-ImI was not impacted by the linker in the dimer. Comparison between the root mean square deviation (RMSD) for α-ImI as a monomer and a dimer showed that the conformation of α-ImI maintained stable over the MD simulations (Figure 3A). Indeed, polyethylene glycol (PEG), as the main component of the linker, provided a spacer segment to vary flexibility, length, and accessibility of α-ImI [33].

The secondary structure of each α-ImI in the 2×ImI-dendrimer was maintained. Comparing the secondary structure of each α-ImI in the 2×ImI-dendrimer in the last 20 ns of the MD simulation and the two α-ImI monomers, we found that the linker had no significant effect on the secondary structure of α-ImI, especially in the last 10 ns (Appendix A). Alignment of the 8 frames extracted from the last 10 ns MD with even time intervals with the α-ImI initial frame revealed that the secondary structure of each α-ImI in the 2×ImI-dendrimer was not significantly changed (Figure 3B). Together, the linker was flexible, and it did not impact the conformation of α-ImI in the dendrimeric form. Each α-ImI in the dimer was structurally intact and free to interact with hα7 nAChR independently.

#### 2.2.2. The Opening of the C-Loop

A peripheral loop in nAChR (Figure 4A), called the C-loop, behaves as a ligand-triggering lid, effectively closing and opening the ligand-binding pocket [40]. Given that the measurement of the C-loop opening is an indicator for the characterization of the effect of the ligands acting on nAChRs [41,42], such measurements have been made to characterize the conformational change of the binding site of the 2×ImI-dendrimer. The results show that the opening of the C-loop in each binding pocket was maintained at about 19Å throughout the whole MD simulation, which was comparable to the values from our previous study [39] (Figure 4B), indicating that the linker between the two α-ImI of the 2×ImI-dendrimer neither entered the binding pocket nor participated in the interaction with the residues at the binding site.

#### 2.2.3. Interacting with hα7 nAChR

In the design of the 2×ImI-dendrimer, the PEG linker not only links the two α-ImI, but also enhances the dendrimer solubility. From the above analysis, we confirmed that the linker was free from the binding site, neither affecting the interaction of α-ImI to the receptor nor interacting with the residues of the binding sites. We were interested in the conformational states of the linker in the MD simulation. Here, we extracted 25 conformations from 1 to 50 ns and 10 conformations in the first 1 ns with even time intervals to observe the conformational change of the linker. Through cluster analysis, we found that the dynamics of the linker evolved from the non-equilibrated and fluctuated state (Appendix A) to the equilibrated and relevantly stable state (Figure 5A). At the beginning of the MD simulation, the linker showed substantial fluctuation from 0 to 14 ns, whereas its flexibility gradually decreased from 14 to 34 ns. Afterward, the linker showed a certain degree of stability (Figure 5A, orange and blue rectangle). The improved stability for the linker originated from its interactions with the region of the receptor between the two adjacent subunits.

Being stabilized by the local residues, the left and right (Figure 5A) fragments of the linker maintained regular curving conformation in the MD simulation. To study the specific role played by the two parts of the linker, we performed cluster analysis to identify the most populated conformations and used these to identify possible interaction partners. According to the above conclusion, the linker reached equilibrium at 34–50 ns, and during this period, we extracted frames with 1 ns intervals to perform a cluster analysis and analyzed the linker conformation change to identify possible interaction partners (Appendix A). Overall, conformations and interactions formed by the right half of linker (UNK8) were significantly more stable than those of UNK9. For UNK8, the U-shape was relatively stable, especially at the last 5ns. UNK9, however, was in a less stable state.

As shown by Appendix A, the linker was most stable in the last 5 ns, and we extracted the last frame to analyze the interactions in detail between the linker and hα7 nAChR. The two-dimensional (2D) interaction diagrams (Appendix A) that were obtained using Schrödinger (Maestro, Version 9.0) showed comprehensive interactions between the linker and the receptor. The three-dimensional (3D) interaction diagrams (Figure 5B,C) of these main interactions showed that the oxygen atoms of UNK8 formed three hydrogen bonds with the three hydrogen atoms of the nitrogen atom on the side chain of K143. Three oxygen atoms surrounded the nitrogen atoms on the side chain of K143 to present a “U” shape, forming an extremely stable hydrogen-bond network. On the other side, UNK9 formed two hydrogen bonds with E189 and C190, respectively. Distance analysis (Appendix A) indicated that these hydrogen bonds were stable in the MD simulation. Overall, our simulation results suggested that the linker was directly involved in interactions with the receptor but without interference to the interactions between α-ImI and hα7 nAChR.

### 2.3. Binding Energy Calculation and Decomposition

In general, the molecular mechanics–generalized Born surface area (MMGB/SA) method is able to give more accurate predictions of the ligand binding affinity than the molecular mechanics–Poisson–Boltzmann surface area (MMPB/SA) method because of the approximations introduced in the MMGB/SA method [39,43]. Indeed, the MMGB/SA performed better than MMPB/SA in several studies [44,45,46]. Here, the MMGB/SA was used to explain why the 2×ImI-dendrimer could significantly enhance the binding free energy in comparison with the native α-ImI monomer.

At the thermodynamic level, the binding affinity was determined by the magnitude of the Gibbs free energy (ΔG). Our calculation results indicate that the binding affinity of the 2×ImI-dendrimer is stronger than that of the α-ImI monomer (Table 1), which is consistent with the experimental results [33]. Each α-ImI of the 2×ImI-dendrimer can be accessible at the hα7 nAChR binding site and collectively cause a multivalency effect. Unlike the monomers, the multivalent α-ImI density at the surface increases, which can strengthen ligand receptor binding probability and improve the targeting of attached components [47].

ΔG is a function of only two terms, the enthalpy (ΔH) and the entropy (ΔS). The influence of enthalpy change (ΔH) can be from dehydration of polar groups of ligands and receptors, hydrogen bond and Van der Waals interactions, etc. As the enthalpy and entropy contribute to the binding energy in an additive fashion (ΔG = ΔH − TΔS), it is clear that reduced enthalpy is beneficial to increase the binding free energy. The ΔH of the 2×ImI-dendrimer was approximately 3-fold lower than its monomeric counterparts (Table 1), suggesting that the interaction between the 2×ImI-dendrimer and hα7 nAChR is stronger than that of the monomer. According to the binding mode (Figure 5B,C), the linker in the 2×ImI-dendrimer also contributed substantially to its enthalpy. Overall, the linker and dual α-ImI jointly prompted favorable enthalpy.

Although the enthalpy of the 2×ImI-dendrimer was approximately 3-fold lower than its monomeric counterparts, the entropy loss (ΔS) of the dimer was 3-fold more than that of the monomer (Table 1). The conformational entropy change usually reflects a loss of conformational degrees of freedom between the ligands and the protein. Consequently, in terms of the loss of conformational degrees of freedom, the loss of the 2×ImI-dendrimer was much greater than that of the monomer. The reason might lie in the PEG spacer units, which is part of the linker. It was flexible prior to binding with the receptor, whereas its conformational constraint increased substantially when the linker interacted with the receptor (Figure 5B,C), therefore, making it an unfavorable term.

In a strategy to obtain the contribution of the linker to α-ImI dimer binding affinity, the binding energy contribution from UNK8 and UNK9 was calculated using MMGB/SA. The calculation results show that the linker made a beneficial contribution to the binding energy. Interestingly, the energy contribution of UNK8 was higher than that of UNK9 (Table 2), indicating that the two parts (UNK8 and UNK9) play different roles in binding with hα7 nAChR. These calculated results are consistent with the results given by the binding mode analysis. The cartoon diagram of the 2×ImI-dendrimer/hα7-nAChR is shown in Figure 6A. The linker had two main effects on binding energy, adverse entropy contribution, and beneficial enthalpy contribution. If the linker was too short, only one α-ImI of the 2×ImI-dendrimer’ could bind to hα7 nAChR, and the linker could not closely interact with hα7 nAChR (Figure 6B). Consequently, the ΔH’ and the ΔS’ both decreased compared with those of the 2×ImI-dendrimer with a reasonable length of linker. Under these circumstances, the potency of the 2×ImI-dendrimer’ was comparable to that of the monomer (ΔG’ = ΔG). If the linker was too long, both α-ImI of the 2×ImI-dendrimer’’ could simultaneously bind to hα7 nAChR, whereas the multivalency effects were significantly decreased due to the overly long linker (Figure 6C). Thus, compared with the 2×ImI-dendrimer with a reasonable length of linker, the ΔH’’ either decreased or did not change, whereas the ΔS’’ decreased. In this case, the potency of the 2×ImI-dendrimer’’ was comparable to that of the monomer or the 2×ImI-dendrimer (ΔG’’ = ΔG).

Overall, our computational studies suggest that the 2×ImI-dendrimer interaction with hα7 nAChR is enthalpy-driven binding and an appropriate linker is necessary to obtain full activity. If the selected linker is too long or too short, one can expect an increase/decrease in the flexibility of the linker potentially affecting the bioactivity. Our findings provide guidance for the design of functional peptide dendrimers and suggest that the linker length, flexibility, and hydrophilicity (e.g., with an appropriate number of hydrogen bond acceptors or donors) should be carefully considered.

## 3. Conclusions

In this study, we established a computational method for the simulation of the interactions between the 2×ImI-dendrimer and hα7 nAChR and statistically analyzed the relationship between the linker, α-ImI, and hα7 nAChR. We found that the 2×ImI-dendrimer maintains its tertiary peptide structure and is free to interact with hα7 nAChR. There were several critical interactions between the linker and hα7 nAChR, which suggested that the multivalent effects are closely related to the linker length and flexibility.

In summary, such a computational method is not only useful to explain the potency of conotoxin dendrimers, but also for the rational design of other peptide dendrimers and small molecular copolymers to improve their potency and specificity. Our work will provide guidance for the design of other dendrimers used as drug/gene carriers, macromolecular drugs, and molecular probes.

## 4. Materials and Methods

### 4.1. α-ImI/hα7-nAChR Complex

In the absence of a crystal structure of hα7 nAChR, hα7 nAChR ligand binding domain (LBD) combined with α-ImI was modeled using a comparative approach [39,48,49,50]. In this study, the structure of AChBP in complex with α-ImI (PDB code: 2c9t) was employed as a structural template to orient the five α7 subunits in the pentamer. The orientations of the side chains were modeled according to the extracellular domain of the α1 nAChR subunit (PDB code: 2qc1). The multiple sequence alignment between AChBP, α-ImI/hα7-nAChR, and the α1 nAChR subunit was generated using MUSCLE [51] and manually adjusted based on structural superimpositions of the AChBP and α1 nAChR subunit. MODELLER (Version 9v14) [52] was then employed to generate 100 3D structural models of the α-ImI/hα7-nAChR complex. The model selected according to the discrete optimized protein energy (DOPE) score [53] was analyzed using MolProbity [54], and more than 94% of residues were in the favorable region of the Ramachandran plot, which is acceptable for a comparative model [53].

### 4.2. 2×ImI-dendrimer/hα7-nAChR Complex

Appendix A illustrates the procedures of the methodology of building the 2×ImI-dendrimer/hα7-nAChR model. In the first stage, UNK8 and UNK9 were constructed by Discovery Studio 3.5 and ChemDraw, and the peptide fraction was generated using the sequence command included in xLEaP. Afterwards, Antechamber and Amber were used to generate parameters for the UNK8/UNK9 and peptide segments, respectively. Next, all the parameters of UNK8/UNK9, UNK8/UNK9.PDB, and the α-ImI.PDB were loaded into xLEaP. The 2×ImI-dendrimer and 2×ImI-dendrimer/hα7-nAChR (one 2×ImI-dendrimer with five α7 subunits) were constructed successively in the editing window of xLEaP. Clearing both tasks, the MD simulation was carried out.

#### 4.2.1. The Building of the 2×ImI-dendrimer

The structure of the linker consisted of two PEG spacer units (UNK8 and UNK9) and a peptide fragment (Lys-Gly-(Arg)4-Gly) (Figure 7). The molecule building module of Discovery Studio 3.5 was used to generate UNK8/UNK9 at the N terminal of α-ImI in the two adjacent binding pockets of hα7 nAChR. The locations of both ends of the linker were determined based on the two N terminals of α-ImI. The rough conformation of the linker was determined based on the concave–convex degree of the hα7 nAChR surface between the two adjacent binding sites. Then, the linker was positioned on the surface of the receptor by carefully checking the heavy atoms distances between the linker and the receptor to avoid atom collisions.

The forcefield used by Discovery Studio 3.5 to construct the linker was CHARMM, which was incompatible with the forcefield used, AMBER. ChemDraw was used here to convert the CHARMM PDB format structures into a PDB format that was compatible and readable with Antechamber. First, ChemDraw 2D was used to construct the two-dimensional structure of UNK8, and then, ChemDraw 3D was used to generate the three-dimensional structure (named UNK8.PDB). UNK8.PDB with modified atomic names can be imported into Antechamber to generate parameters. The procedure for generating parameters for UNK9.PDB was the same as that for UNK8.PDB.

We used AMBER forcefield ff14SB for the peptide fraction, and general AMBER force field GAFF parameters for UNK8 and UNK9. Parameters for UNK8 and UNK9 (Appendix A) were generated using the Antechamber module embedded in AmberTools in AMBER16 package [55]. Atom partial charges for UNK8 and UNK9 were produced using the R.E.D Tools [56].

#### 4.2.2. Molecular Dynamics Simulation

The 2×ImI-dendrimer/hα7-nAChR complex that we simulated contained one α-ImI-dendrimer and five α7 subunits. The 2×ImI-dendrimer/hα7-nAChR model was solvated in a cubic box with 49,498 water molecules, and 13 Na+ were added to simulate and neutralize the system. The ff14SB forcefield [57] was applied to the protein and peptide, and the GAFF was used for the linker region. Once we built the whole system in xLEaP of AMBER16, minimization before the MD simulation was performed to remove the Van der Waals contacts between the α-ImI dendrimer and hα7 nAChR. The first was constrained optimization, which was the steepest descent method optimization of 2000 steps and the conjugate gradient method optimization of 3000 steps, and the solute binding force was 100 kcal mol. After completing the first round of energy optimization, the binding forced was removed from the solute molecules, unconstrained optimization was performed, and the entire system was optimized using the same parameters as above. The MD simulations were carried out after minimization using methods as described previously [58]. Additionally, MD simulations of two α-ImI monomers bound with five α7 subunits were also performed.

#### 4.2.3. Binding Energy Calculations

MMGB/SA [59] was applied to calculate the binding affinities of the 2×ImI-dendrimer and the parts of the linker (UNK8 and UNK9) against hα7 nAChR. The energies were averaged on the 50 frames extracted from the last 10 ns of the MD simulation.

The values of the binding free energy (ΔG binding) for each model were calculated based on the following equation:ΔG_binding_ = G_complex_ − G_ligand_ − G_receptor_.(1)

The free energy can be decomposed into three components (complex, receptor, and ligand):G = <G_solute_> + <G_epol_> + <GSA>,(2)
where G_solute_ is the solute Gibbs free energy, G_epol_ represents the polar contribution to the solvation energy, and GSA represents non-polar contribution to the solvation energy. The internal dielectric and external dielectric constants were set to 2.0 and 80.0, respectively. A probe radius of 1.4 Å, a grid spacing of 0.5 Å, and ionic strength of 0.15 mol/L were set up for the calculations. Other parameters of the energy calculation were described previously in [39].

The entropy can be divided into solvation and configuration entropy. The configuration entropy is a measure of the entropy in the solute (protein–ligand complex) in solution and is the only entropy element calculated explicitly [60]. Oehme et al. calculated the entropy used the NMODE module to explain the effect of entropy on binding free energies [61]. They found that the entropy estimates correlate with the solvent-accessible surface area (SASA) of the ligands. In this work, estimates of the entropy contribution to the binding free energy (-TΔS) used the NMODE module of AMBER. The calculation condition was limited to structures every 100 ps (200 snapshots over the 5 ns trajectory) from 34th ns to 50th ns. To find a balance between accuracy and efficiency, 12 Å was kept, and the other less important residues were truncated. We used a notation of ΔG’’’ to indicate binding energies without the inclusion of entropy, whereas ΔG refers to the binding energy inclusive of entropy.

## Figures and Tables

**Figure 1 marinedrugs-17-00303-f001:**
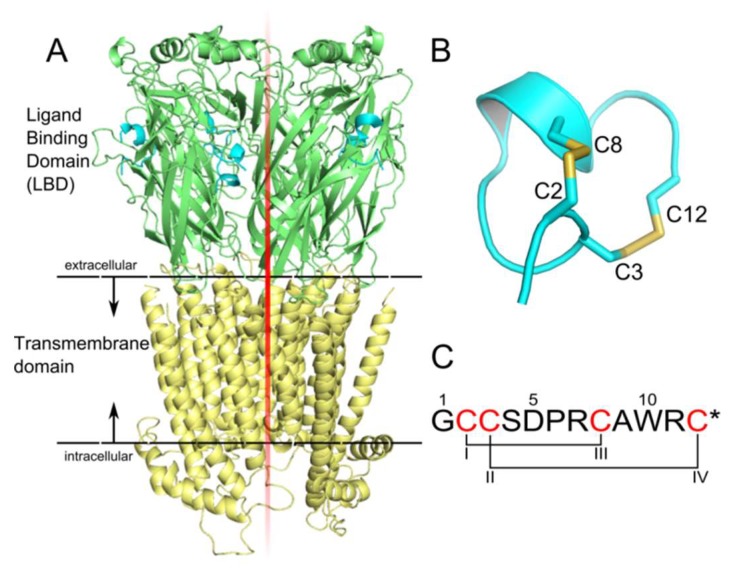
Structures of a nicotinic acetylcholine receptor (nAChR) and α-ImI. (**A**) hα7 nAChR bound with α-ImI. nAChRs are ligand-gated ion channels. The structure of nAChR consists of a ligand binding domain (green), a transmembrane domain (yellow), and an intracellular domain (purple). Five α-ImI (light blue) bound to the acetylcholine binding sites. The α-ImI/hα7-nAChR complex structure was built using the crystal structure of α1 subunit (Protein data bank (PDB) code: 2qc1) and AChBP/ImI (PDB code: 2c9t) as templates. (**B**,**C**) α-ImI comprises 12 residues and is C-terminal amidated (indicated by *). The structure features a short α-helix and two disulfide bonds that link cysteines I–III and II–IV. The cysteines are highlighted in red.

**Figure 2 marinedrugs-17-00303-f002:**
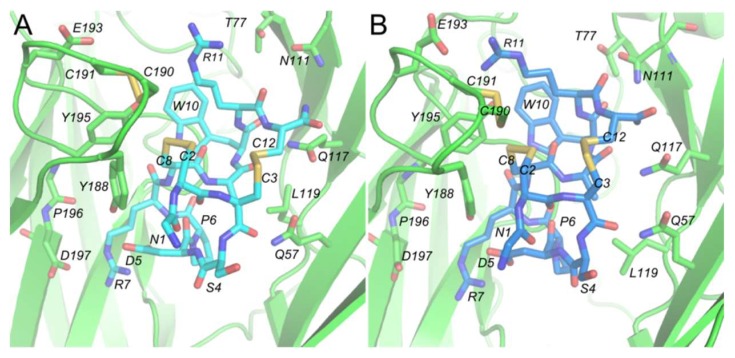
Binding mode of the α-ImI monomer and dimer. (**A**) the binding mode of the α-ImI monomer (light blue). (**B**) the binding mode of α-ImI of the 2×ImI-dendrimer in the same pocket as (**A**) (blue). The hα7 nAChR is shown in cartoon, while α-ImI and the residues of the binding site are shown in stick form.

**Figure 3 marinedrugs-17-00303-f003:**
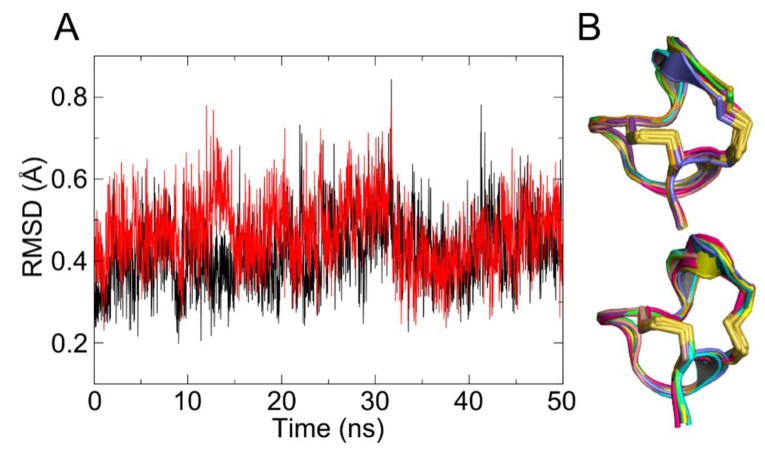
The conformation of the two α-ImI in the dimer. (**A**) Evolution of the root mean square deviation (RMSD) for each α-ImI in the dimer in the 50 ns molecular dynamics (MD) simulations. (**B**) Superposition diagram of the averagely extracted 8 frames (in varied colors) from the last 10 ns and the α-ImI initial frame (in blue).

**Figure 4 marinedrugs-17-00303-f004:**
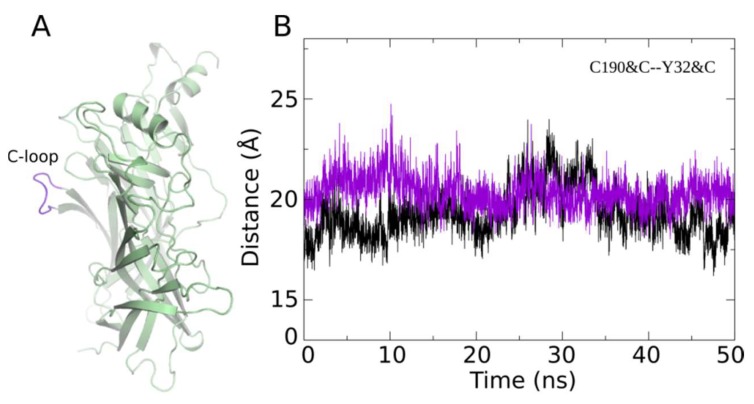
The opening of the C-loop in two adjacent binding sites. (**A**) Side view of the hα7 nAChR subunit. A peripheral loop highlighted in purple represents the C-loop. (**B**) C-loop opening measurement in the MD simulation. It was measured by calculating the distance between the CA atom of C190 and the CA atom of Y32 in hα7 nAChR over time. Purple and black represent the distance variation of two different binding sites, respectively.

**Figure 5 marinedrugs-17-00303-f005:**
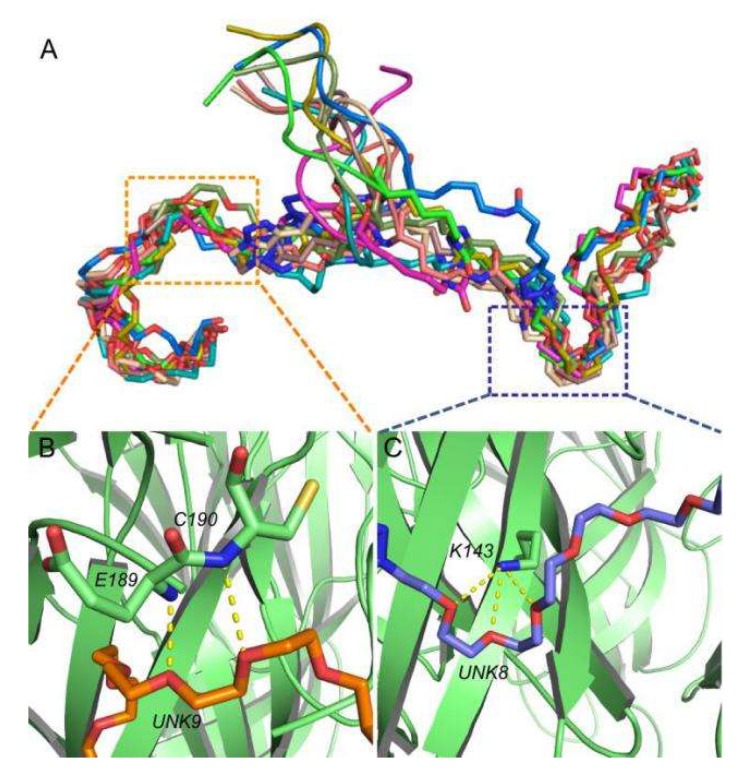
Conformation and binding modes of the linker. (**A**) The average extraction frames from 34th ns to 50th ns. Linker was divided into two parts, named UNK8 and UNK9 (Figure 7). (**B**) Binding mode of the left half of linker (UNK9) at hα7-nAChR; (**C**) Binding mode of the right half of linker (UNK8) at hα7-nAChR. hα7 nAChR is shown in green, and the two parts of the linkers are colored in orange (UNK9) and purple (UNK8), respectively.

**Figure 6 marinedrugs-17-00303-f006:**
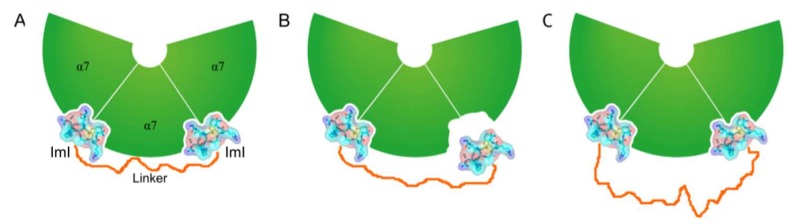
The 2×ImI-dendrimer/hα7-nAChR model with different lengths of linkers. hα7 nAChR, α-ImI, and the linker are labeled. (**A**) The 2×ImI-dendrimer model with a linker of reasonable length. Its enthalpy change, entropy change, and binding energy were expressed as ΔH, ΔS, and ΔG. (**B**) The 2×ImI-dendrimer model with a linker that was too short (2×ImI-dendrimer’). Its enthalpy change, entropy change, and binding energy were expressed as ΔH’, ΔS’, and ΔG’. (**C**) The 2×ImI-dendrimer model with an overly long linker (2×ImI-dendrimer’’). Its enthalpy change, entropy change, and binding energy were expressed as ΔH’’, ΔS’’, and ΔG’’.

**Figure 7 marinedrugs-17-00303-f007:**
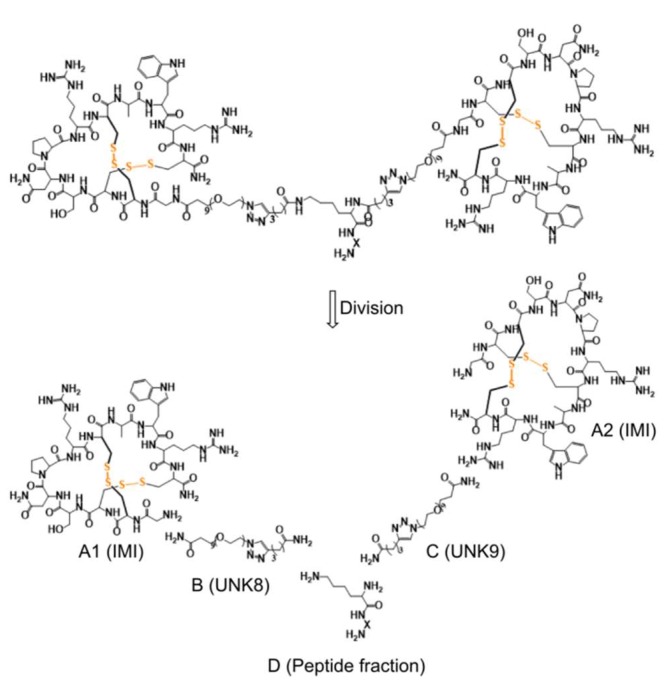
The structure diagram of the ligand and the linker. (A1) and (A1) represent α-ImI. (B) and (C) are the polyethylene glycol (PEG) spacer units, which were named UNK8 and UNK9 respectively. (D) represents the peptide fraction, and the “X” represents 6 amino acid residues (Gly-Arg-Arg-Arg-Arg-Gly).

**Table 1 marinedrugs-17-00303-t001:** Energy contribution of the α-ImI monomer and dimer.

Ligands	ΔG (Kcal/mol)	ΔH (Kcal/mol)	TΔS (Kcal/mol)
α-ImI	−30.68	−73.70	−43.02
2×ImI-dendrimer	−78.46	−233.85	−155.39

**Table 2 marinedrugs-17-00303-t002:** Energy contribution of two segments of the linker.

Ligands	ΔG’’’ (Kcal/mol)
UNK8	−22.32
UNK9	−13.12

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
