# Peer review of "Blockade of Human α7 Nicotinic Acetylcholine Receptor by α-Conotoxin ImI Dendrimer: Insight from Computational Simulations"

_marinedrugs, 2019, doi:10.3390/md17050303_

Reviewer 1 Report

A general observation is that the study contains new and interesting information on the molecular interactions between α-ImI-dendrimers (2x and 4x) and hα7-nAChR. There is only need for minor improvements which will make the manuscript more comprehensive and reader friendly. Furthermore, the use of English needs some improvement; although not compromising text comprehension, there are numerous grammatical and syntax errors that need to be corrected upon revision, so the authors are kindly advised to have their manuscript checked by a native English speaker or a relevant professional service. Detailed comments follow:

1. Introduction

- Page 2, line 43: Please place “Conus” in italics, as it refers to a genus scientific name.

2. Results and Discussion:

- Sections 2.1, 2.2.1 and 2.2.2: The structure in these sections seems like discussing the results before they are presented. Please reorganize these sections, to the best possible degree, so that results of the study are presented first and then comparisons with existing bibliography or previous experiments of the group can follow.

- Page 3, line 84: Please explain what “MD” stands for at this first instance in the text.

- Page 3, line 97: Please explain what “PEG” stands for at this first instance in the text.

- Page 7, line 208: (a) Please replace “both” with “all” (three parameters indicated). (b) Please replace “hydrophily” with the correct term “hydrophilicity”.

- Page 7, lines 208-209: What does the “ex” mean in the phrase “ex with appropriate number of hydrogen bond acceptors or donors”?

3. Materials and Methods:

- Page 9, line 253: Please correct “ChenDraw” to “ChemDraw”.

4. Conclusions

It is suggested that this part is placed before the Materials and Methods section, after the Results and Discussion, where it fits better.

- Page 11, lines 311-313: Please provide some more details on the potential impact and the usefulness of the study, mostly in terms of the benefits for the design of molecules applicable to the pharmaceutical sector.

Author Response

Response to Reviewer 1 Comments

Point 1: - Page 2, line 43: Please place “Conus” in italics, as it refers to a genus scientific name.

Response 1: Thanks for the reviewer’s remind. We have placed “Conus” in italics. (- Page 1, line 41)

Point 2: - Sections 2.1, 2.2.1 and 2.2.2: The structure in these sections seems like discussing the results before they are presented. Please reorganize these sections, to the best possible degree, so that results of the study are presented first and then comparisons with existing bibliography or previous experiments of the group can follow.

Response 2: According to your suggestion, we reorganize these sections:

2.1. Comparing the binding mode of α-ImI monomer and α-ImI dimer at hα7 nAChR

α-ImI monomer and α-ImI dimer possessed similar binding modes at hα7 nAChR (Figure 2, A, B, S1). The critical residues conferring α-ImI specificity for the hα7 nAChR are clustered at one site of the α-ImI, which is consistent with previous NMR structural study of α-ImI [37]. Comparing the conformations of α-ImI before and after MD (Molecular Dynamics) simulations, it was found that the conformations of D5, R7 and W10 were well maintained, while the R11 had a slight orientation change (Figure 2, A, B). Results from previous mutagenesis study and computational modeling suggested that the D5, R7 and W10 in α-ImI contributed most to the binding affinity of α-ImI [38,39], which are in line with the results from our modeling of α-ImI/hα7-nAChR monomers and 2×ImI-dendrimer/hα7-nAChR in current study (Figure 2, A, B). The overall results suggest that conformations for these critical residues in each α-ImI of the dimer were maintained after MD simulations, and the linker had minor effects on α-ImI binding to the hα7 nAChR.

2.2.1. Conformation of α-ImI in the 2×ImI-dendrimer

Conformation of the α-ImI was not impacted by the linker in the dimer. Comparison between the RMSD for α-ImI as a monomer and a dimer showed that the conformation of the α-ImI maintained stable over the MD simulations (Figure 3A). Indeed, PEG (Polyethylene glycol) as the main component of the linker, provided a spacer segment to vary flexibility, length, and accessibility of the α-ImI [33].

The secondary structure of each α-ImI in 2×ImI-dendrimer was maintained. Comparing the secondary structure of each α-ImI in 2×ImI-dendrimer in the last 20 ns MD and two α-ImI monomers, we found that the linker has no significant effect on the secondary structure of the α-ImI, especially in the last 10 ns (Figure S2, A, B). Alignment of the 8 frames extracted from the last 10 ns MD with even time intervals with the α-ImI initial frame revealed that the secondary structure of each α-ImI in 2×ImI-dendrimer was no significantly changed (Figure 3B). Together, the linker was flexible and it did not impact the conformation of α-ImI in the dendrimeric form. Each α-ImI in the dimer is structurally intact and free to interact with hα7 nAChR independently.

2.2.2. The opening of the C-loop

A peripheral loop in nAChR (Figure 4A), called the C-loop, behaves as a ligand-triggering lid, effectively closing and opening the ligand-binding pocket [40]. Given that the measurement of C-loop opening as an indicator for characterization of the effect of the ligands acting on nAChRs [41,42], such measurements have been made to characterize the conformational change of the binding site with 2×ImI-dendrimer bound. The result showed that the opening of the C-loop in each binding pocket was maintained at about 19Å in the whole MD, which was comparable to the values from our previous studies [39] (Figure 4B), indicating that the linker between the two α-ImI of the 2×ImI-dendrimer neither entered the binding pocket, nor did it participate in the interaction with residues at the binding site.

Point 3: - Page 3, line 84: Please explain what “MD” stands for at this first instance in the text.

Response 3: The “MD” stands for “Molecular Dynamics”. Explanations have been made in the article. (- Page 3, line 78)

Point 4: - Page 3, line 97: Please explain what “PEG” stands for at this first instance in the text.

Response 4: The “PEG” stands for “Polyethylene glycol”. Explanations have been made in the article. (- Page 3, line 95)

Point 5: - Page 7, line 208: (a) Please replace “both” with “all” (three parameters indicated). (b) Please replace “hydrophily” with the correct term “hydrophilicity”.

Response 5: Thanks for corrections. Errors have been corrected. (- Page 7, line 216)

Point 6: - Page 7, lines 208-209: What does the “ex” mean in the phrase “ex with appropriate number of hydrogen bond acceptors or donors”?

Response 6: We made a mistake here. It should be “e.g”. We have replaced “ex” with “e.g.” in the main text. (- Page 7, line 216)

Point 7: - Page 9, line 253: Please correct “ChenDraw” to “ChemDraw”.

Response 7: Thanks for your remind. Errors have been corrected. (- Page 8, line 256)

Point 8: It is suggested that this part is placed before the Materials and Methods section, after the Results and Discussion, where it fits better.

Response 8: Thanks for your suggestion. We have placed the “Conclusions” before the Materials and Methods section, after the Results and Discussion. Indeed, this arrangement makes the structure of the article seem to be better.

Point 9: - Page 11, lines 311-313: Please provide some more details on the potential impact and the usefulness of the study, mostly in terms of the benefits for the design of molecules applicable to the pharmaceutical sector.

Response 9: We have provided some more details on the potential impact and the usefulness of the study (- Page 8, line 229-239), as following:

In this study, we established a computational method for simulation of the interactions between the 2×ImI-dendrimer and hα7 nAChR, and statistically analyzed the relationship between the linker, α-ImI and hα7 nAChR. We found that the 2×ImI-dendrimer maintain their tertiary peptide structure and are free to interact with hα7 nAChR. There are several critical interactions between the linker and the hα7 nAChR, which suggests that the multivalent effects are closely related to the linker length and flexibility.

In summary, such a computational method can not only be used to explain the potency of conotoxin dendrimers, but also useful for the rational design of other peptide dendrimers and small molecular copolymers for improving their potency and specificity. Our work will provide guidance for the design of other dendrimers used as drug/gene carries, macromolecular drugs and molecular probes.

Reviewer 2 Report

Could the authors please include the parameters for the linker in supplementary details.

Also, could the authors please provide more details regarding construction of the dendrimer/alpha7 complexes? Specifically, while it is understood that homology modelling was used to construct the alpha7/ImI complex, and Discovery Studio was used to build the C-terminal linker, what methods were used to determine the initial conformations of the linkers? Was there a geometry optimization or conformer search protocol that was employed to produce a starting guess structure? If so, what forcefield was used for these preparatory steps using Discovery Studio, prior to parameter determination with Antechamber and simulation using Amber?

Apologies if I’ve misunderstood, but it doesn’t seem to be mentioned whether the ImI-alpha7 complexes included all five subunits (and ImIs); all of the analyses seem to suggest only one dimer/monomer was analysed. Did the simulation include all five subunits, with 2 dendrimeric ImI dimers; or just three alpha7 subunits and a single dimer? If the latter, some comment should be made regarding potential differences in the dynamics between a full pentamer versus a trimer.

Line 101: “….revealed that the secondary structure of each α-ImI in 2×ImI-dendrimer was maintained”. Some evidence should be shown in supplementary (eg. DSSP plots).

Line 127: In this section, interactions between the linker and receptor were described. The last frame was described, in which H-bonds were identified and subsequently examined in more detail (Figure S3A and B). But a more comprehensive study could readily be performed using eg. 2D interaction diagrams to identify further potential H-bonding partners, or other stable contacts, which could also play a role in maintaining the linker’s conformation.

The authors should also perform a cluster analysis to identify the most populated conformations, and use these to identify possible interaction partners, rather than (or in addition to) using a single (final) frame.

Section 2.4: While a 4x dendrimer was constructed, no further simulation work was done on it. Therefore, some estimate of binding affinity using MMGBSA (perhaps even just on the starting structure) is needed to justify discussing the possible relative potency of the 4x dendrimer compared to the 2x dendrimer that was actually simulated in this work. However, this would belong better in another manuscript, and given the conclusion in this paper that the linker in the 2x dendrimer was probably insufficiently long, I feel that this section is not needed, as it seems to weaken the other findings in this paper.

Related to the above point, in the Abstract, it is stated that “Additionally, a 4×ImI-dendrimer/hα7-nAChR model was also built, and we found that its comparable potency to 2×ImI-dendrimer might be originated from its insufficient linker length.”. This sentence is unclear, and probably unnecessary.

Minor typographical errors:

Line 253: “ChenDraw” = “ChemDraw”

Line 254: “After this, Antechamber and Amber were used to generate two parameters, respectively”; should this be  “…were used to generate parameters for the UNK8/UNK9 and peptide/protein segments, respectively”?

Line 270: “There are lacking of forcefield parameters…” = “There is a present lack of forcefield parameters….”

Line 289: “Gsolute”; solute should be subscript

This list is not exhaustive, and the authors should carefully proofread the manuscript for further errors and inconsistencies.

Author Response

Response to Reviewer 2 Comments

Point 1: Could the authors please include the parameters for the linker in supplementary details.

Response 1: The parameters for the calculated partial charges of the linker have been attached in supplementary (Table S1). 

Point 2: Also, could the authors please provide more details regarding construction of the dendrimer/alpha7 complexes? Specifically, while it is understood that homology modelling was used to construct the alpha7/ImI complex, and Discovery Studio was used to build the C-terminal linker, what methods were used to determine the initial conformations of the linkers? Was there a geometry optimization or conformer search protocol that was employed to produce a starting guess structure? If so, what forcefield was used for these preparatory steps using Discovery Studio, prior to parameter determination with Antechamber and simulation using Amber?

Response 2: We have provided more details regarding construction of the dendrimer/alpha7 complexes in the article (4.1. , 4.2. and 4.2.1.).

1.1 The detailed method of homology modelling to get alpha7/ImI complex are as follows:

In the absence of the crystal structure of the hα7 nAChR, the hα7 nAChR LBD bound with α-ImI was modeled using a comparative approach. In this study, the structure of AChBP in complex with ImI (PDB code: 2c9t) were employed as structural templates to orient the five α7 subunits in the pentamer. The orientations of the side chains were modeled according to the α1 nAChR subunit (PDB code: 2qc1). The multiple sequence alignment between AChBP, α-ImI/hα7 nAChR and α1 nAChR subunit was generated using MUSCLE, and manually adjusted based on structural superimpositions of the AChBP and α1 nAChR subunit. MODELLER (Version 9v14) was then employed to generate 100 3D structural models of the α-ImI/hα7-nAChR complex. The model selected according to the DOPE score was analyzed using MolProbity and more than 94% residues were in the favorable region of the Ramachandran plot, which is acceptable for a comparative model.

2.2 The method to determine the initial conformations of the linker

The location of both ends of the linker was determined by the N-termini of the α-ImI in the two adjacent binding pockets. The initial conformation of the linker was determined based on the concave-convex degree of the hα7 nAChR surface between the two binding sites. Then linker was positioned on the surface of the receptor by carefully checking the heavy atoms distances between the linker and the receptor for avoiding of atom collisions.

2.3 There was no geometry optimization or conformer search protocol employed to produce a starting guess structure, because the conformation of the linker needs to be in a rough “V” shape so that it is possible to connect the α-ImI to both ends and not to collide with the receptor. If optimized, the linker conformation at this time will not be conducive to building the α-ImI-dendrimer/hα7-nAChR complex. We used ChemDraw 3D to optimize the linker to build the α-ImI-dendrimer/hα7-nAChR complex, but the optimized linker was in an irregular state and cannot be used to build a dimer. But prior to parameter determination with Antechamber and simulation using Amber, the CHRAMN forcefield was used for built the linker in Discovery Studio.

The revised related sections in article are as follows:

4.1. α-ImI/hα7-nAChR complex

In the absence of a crystal structure of the hα7 nAChR, the hα7 nAChR LBD combined with α-ImI was modeled using a comparative approach [39,48–50]. In this study, the structure of AChBP in complex with α-ImI (PDB code: 2c9t) was employed as structural templates to orient the five α7 subunits in the pentamer. The orientations of the side chains were modeled according to the extracellular domain of the α1 nAChR subunit (PDB code: 2qc1). The multiple sequence alignment between AChBP, α-ImI/hα7-nAChR and α1 nAChR subunit was generated using MUSCLE [51], and manually adjusted based on structural superimpositions of the AChBP and α1 nAChR subunit. MODELLER (Version 9v14) [52] was then employed to generate 100 3D structural models of the α-ImI/hα7-nAChR complex. The model selected according to the DOPE score [53] was analyzed using MolProbity [54] and more than 94% residues were in the favorable region of the Ramachandran plot, which is acceptable for a comparative model [53].

4.2. 2×ImI-dendrimer/hα7-nAChR complex

Figure S5 illustrates the procedures of the methodology of building the 2×ImI-dendrimer/hα7-nAChR model. In the first stage, the UNK8 and UNK9 were constructed by Discovery Studio 3.5 and ChemDraw and the peptide fraction was generated using the sequence command included in xLEaP. Afterwards, Antechamber and Amber were used to generate parameters for the UNK8/UNK9 and peptide segments, respectively. Next, all the parameters of UNK8/UNK9, UNK8/UNK9.PDB and the α-ImI.PDB were loaded into xLEaP. 2×ImI-dendrimer, 2×ImI-dendrimer/hα7-nAChR (one 2×ImI-dendrimer with five α7 subunits) were constructed successively in the editing window of xLEaP. Clearing both tasks, MD carried out.

4.2.1. The building of the 2×ImI-dendrimer 

The structure of the linker consists of two PEG spacer units (UNK8 and UNK9) and a peptide fragment (Lys-Gly-(Arg)4-Gly) (Figure 8). The molecule building module of Discovery Studio2018SP1 (Biovia, San Diego, CA) was used to generate the UNK8/UNK9 at the N terminal of the α-ImI in the two adjacent binding pocket of hα7 nAChR. Location of both ends of the linker was determined based on the two N terminals of the α-ImI. The rough conformation of the linker was determined based on the concave-convex degree of the hα7 nAChR surface between the two adjacent binding sites. Then the linker was positioned on the surface of the receptor by carefully checking the heavy atoms distances between the linker and the receptor for avoiding of atom collisions.

The forcefield used by Discovery Studio to construct the linker was CHRAMN, which was incompatible with the forcefield used AMBER. ChemDraw was used here to convert the CHRAMN PDB format structures into a PDB format that were compatiable and readable with Antechamber. Firstly, ChemDraw 2D was used to construct the two-dimensional structure of UNK8, and then ChemDraw 3D was used to generate the three-dimensional structure (named UNK8.PDB). The UNK8.PDB with the modified atomic names can be imported into the Antechamber to generate parameters. The procedures of generation parameters for UNK9.PDB was the same as that of the UNK8.PDB.

We used the AMBER forcefield ff14SB for the peptide fraction, and gaff forcefield parameters for the UNK8 and UNK9. Parameters for the UNK8 and UNK9 (Table S) were generated using the Antechamber module embedded in AmberTools in AMBER16 package [55]. Atom partial charges for UNK8 and UNK9 were produced using the R.E.D Tools [56]. 

Point 3: Apologies if I’ve misunderstood, but it doesn’t seem to be mentioned whether the ImI-alpha7 complexes included all five subunits (and ImIs); all of the analyses seem to suggest only one dimer/monomer was analyzed. Did the simulation include all five subunits, with 2 dendrimeric ImI dimers; or just three alpha7 subunits and a single dimer? If the latter, some comment should be made regarding potential differences in the dynamics between a full pentamer versus a trimer.

Response 3: The 2×ImI-dendrimer/hα7-nAChR complex that we simulated contained one α-ImI-dendrimer and five α7 subunits. In addition, MD simulations of two ImI monomer bound with five α7 subunits were also performed. (- Page 10, line 291-291, 302-303)

Point 4: Line 101: “….revealed that the secondary structure of each α-ImI in 2×ImI-dendrimer was maintained”. Some evidence should be shown in supporting information (eg. DSSP plots).

Response 4: Secondary structure analysis for each α-ImI in 2×ImI-dendrimer was performed in VMD, and was shown in supporting information (Figure S2, B). A represents the secondary structure diagram of two ImI monomers. This diagram are as follows:

Point 5: Line 127: In this section, interactions between the linker and receptor were described. The last frame was described, in which H-bonds were identified and subsequently examined in more detail (Figure S3A and B). But a more comprehensive study could readily be performed using eg. 2D interaction diagrams to identify further potential H-bonding partners, or other stable contacts, which could also play a role in maintaining the linker’s conformation.

The authors should also perform a cluster analysis to identify the most populated conformations, and use these to identify possible interaction partners, rather than (or in addition to) using a single (final) frame.

Response 5: 2D interaction diagrams of the linker was shown in supporting information (Figure S5). From the 2D interaction diagrams, the main interactions of the linker with the hα7-nAChR are consistent with the 3D interaction diagrams.

 The linker reached equilibrium at 34-50ns, so we take a conformation every 1 ns between such time span to perform a cluster analysis, and analyzed the linker conformation change and identified possible interaction partners (Figure S4). Overall, conformations and interactions formed by UNK8 were significantly stable than that of the UNK9. For UNK8, the U-shape was relatively stable, especially at the last 5ns. The UNK9, however, was in a less stable state. Figure S4 shows that the linker was most stable in the last 5 ns, and we extracted the last frame to analyze the interactions in detail between the linker and the hα7 nAChR. Cluster analysis diagram of the linker between 34 ns to 50 ns are as follows:

Point 6: Section 2.4: While a 4x dendrimer was constructed, no further simulation work was done on it. Therefore, some estimate of binding affinity using MMGBSA (perhaps even just on the starting structure) is needed to justify discussing the possible relative potency of the 4x dendrimer compared to the 2x dendrimer that was actually simulated in this work. However, this would belong better in another manuscript, and given the conclusion in this paper that the linker in the 2x dendrimer was probably insufficiently long, I feel that this section is not needed, as it seems to weaken the other findings in this paper.

Related to the above point, in the Abstract, it is stated that “Additionally, a 4×ImI-dendrimer/hα7-nAChR model was also built, and we found that its comparable potency to 2×ImI-dendrimer might be originated from its insufficient linker length.”. This sentence is unclear, and probably unnecessary.

Response 6: : According to the reviewer’s suggestion, we thought the 4x dendrimer was redundant to this work. We deleted this section and modified the Abstract.

Point 7: Line 253: “ChenDraw” = “ChemDraw”

Response 7: Thanks for your remind. It was revised. (- Page 8, line 257)

Point 8: “After this, Antechamber and Amber were used to generate two parameters, respectively”; should this be  “…were used to generate parameters for the UNK8/UNK9 and peptide/protein segments, respectively”?

Response 8: Thanks for your remind. According to your suggestion, we have changed the sentence to: Antechamber and Amber were used to generate parameters for the UNK8/UNK9 and peptide segments, respectively. (- Page 8, line 258, 259)

Point 9: Line 270: “There are lacking of forcefield parameters…” = “There is a present lack of forcefield parameters….”

Response 9: Thanks for your remind. Errors have been corrected. (- Page 9, line 285)

Point 9: Line 289: “Gsolute”; solute should be subscript

Response 9: Thanks for your remind. Errors have been corrected. (- Page 10, line 318)

Round  2

Reviewer 2 Report

The authors have addressed my major comments and I recommend publication of the manuscript in its present form. However, I also suggest the authors carefully proofread their revised manuscript, as there are still some apparent typos (eg. should "CHRAMN" be "CHARMM" forcefield in section 4.2.1?).

Author Response

Point 1: However, I also suggest the authors carefully proofread their revised manuscript, as there are still some apparent typos (eg. Should “CHRAMN” be “CHARMM” forcefield in section 4.2.1?).

Response 1: Thanks for your remind. Errors have been corrected. (- Page 9, line 272 and 273) 

Additionally, we also identified other grammar errors, and they were also corrected with the trace maintained.

- Page 2, line 46: Corrected “IC50” to “IC50.

- Page 2, line 55: Corrected “`” to “ .

- Page 2, line 69: Corrected “in silico” to “in silico.

- Page 4, line 122: Corrected “subunits” to “subunit.

- Page 5, line 126: Corrected “a” to “α.

- Page 5, line 150: Corrected “SCHRODINGER” to “Schrödinger (Maestro, version 9.0).

- Page 6, line 162: Corrected “/” to “at.

- Page 7, line 189: Corrected “times” to “fold.

- Page 7, line 213: Corrected “interaction” to “interacted.

- Page 8, line 265: Corrected “Discovery Studio2018SP1 (Biovia, San Diego, CA)” to “Discovery Studio 3.5.

- Page 9, line 272: Corrected “Discovery Studio” to “Discovery Studio 3.5.

- Page 9, line 272 and 273: Corrected “CHRAMN” to “CHARMM.

- Page 9, line 274: Corrected “compatiable” to “compatible.

- Page 9, line 280: Corrected “gaff” to “GAFF.
